# Rhinosinusitis Treatment with Cineole: Patient-Reported Quality of Life Improvements from a Non-Interventional, Pharmacy-Based Survey

**DOI:** 10.3390/medicines10060037

**Published:** 2023-06-19

**Authors:** Nina Werkhäuser, Ursula Pieper-Fürst, Hacer Sahin, Antonia Claas, Ralph Mösges

**Affiliations:** 1Clin Competence Cologne GmbH, Theodor-Heuss-Ring 14, 50668 Cologne, Germany; nina.werkhaeuser@clincompetence.de (N.W.); ursula.pieper-fuerst@clincompetence.de (U.P.-F.); hacer.sahin@clincompetence.de (H.S.); 2Institute of Medical Statistics and Computational Biology (IMSB), Faculty of Medicine, University of Cologne, Kerpener Str. 62, 50937 Cologne, Germany; antonia.claas@clincompetence.de

**Keywords:** rhinosinusitis, cineole, quality of life

## Abstract

Background: Rhinosinusitis is commonly treated with decongestants, analgesics, and local corticosteroids. Phytotherapeutics are also utilised for symptomatic relief, including cineole, the main component of eucalyptus oil. Methods: The current non-interventional, anonymised survey investigated quality of life in participants with rhinosinusitis (with or without additional symptoms of bronchitis) via the German version of a validated quality of life questionnaire (RhinoQol). Overall, 310 subjects administered a cineole preparation (Sinolpan) and 40 subjects applying nasal decongestant were recruited in German pharmacies. Results: Significant improvements in frequency (64.0%), bothersomeness (52.1%), and impact (53.9%) of rhinosinusitis symptoms were reported upon treatment with cineole over a mean treatment period of seven days (*p* < 0.001 each). The overall treatment efficacy of cineole was evaluated as good or very good by 90.0% of the participants, and the quality of life during work or leisure time improved upon treatment. Six (non-serious) possibly related side effects were reported in four participants who were administered cineole. The tolerability of the treatment was assessed as good or very good by 93.9% of the participants. Conclusions: Cineole can be considered as a safe and well-tolerated rhinosinusitis treatment conferring a clear improvement in quality of life outcomes.

## 1. Introduction

Rhinosinusitis is a highly prevalent (6–15% yearly) inflammatory disease of the nose and paranasal sinuses. The disease can be classified as acute or chronic, differentiated by duration (duration ≤ 12 weeks, acute; duration > 12 weeks, chronic) [1]. Typical symptoms of rhinosinusitis are nasal congestion/obstruction/congestion or nasal discharge (anterior/postnasal drip) often accompanied by facial pain/pressure and reduction/anosmia [1]. Acute rhinosinusitis (ARS) is most commonly caused by viruses including rhinovirus, adenovirus, influenza virus, and parainfluenza virus. The literature states that less than 2% of ARS cases are of bacterial cause [2,3]. Although acute rhinosinusitis is mostly self-limiting, it may constitute a significant burden for individuals and greatly impair quality of life [4]. Rhinosinusitis treatment is mainly symptomatic, and several treatments such as decongestants, analgesics, local corticosteroids, and herbal medicines including cineole are recommended [1,5]. Despite the fact that antibiotics are only recommended in patients with symptoms of bacterial rhinosinusitis (fever, purulent discharge, and severe unilateral facial pain), 80% of patients with ARS are prescribed antibiotics [2].

Thereby, rhinosinusitis constitutes for the fifth most common cause of antibiotics prescription in adults [3]. Since antibiotic resistance is a rising concern, the exploration of herbal and alternative preparations has increasing relevance and importance. 1,8-cineole is the main constituent of eucalyptus. It has shown to have anti-inflammatory and antioxidant properties, which mainly function via regulation of the nuclear factor-kappa B (NF-ĸB) and nuclear factor erythroid-2-related factor 2 (Nrf2) pathway [6]. Moreover, 1,8-cineole has shown to be a strong inhibitor of the pro-inflammatory cytokines TNF-α and IL-1β [7]. Ex vivo studies have additionally demonstrated the reduction effect of 1,8-cineole on mucus overproduction. Notably, a reduced number of mucin-filled goblet cells were observed post cineole treatment, which was explained by the significant downregulation of mucin genes (MUC2 and MUC19) [8]. Accordingly, a placebo-controlled study in patients with acute non-purulent rhinosinusitis demonstrated a clinically significant reduction in symptom sum score after both four and seven days of treatment with cineole. Patients reported improvements in classical symptoms, which make up the sum score: headache on bending, sensitivity of trigeminal nerve pressure points, impairment of general condition, nasal obstruction, rhinological secretion, secretion quantity, secretion viscosity, frontal headache, and fever [9]. Another study assessing the reduction of the same symptom score demonstrated the superior efficacy of cineole when compared to an alternative herbal preparation in the treatment of acute viral rhinosinusitis [10]. Furthermore, a placebo-controlled study in acute bronchitis patients showed significant improvement of the bronchitis sum score (parameters include dyspnoea, sputum, frequency of cough, thoracic pain, auscultation, and lung function) upon treatment with cineole [11].

The aim of the current survey was to investigate the quality of life in participants suffering from rhinosinusitis and using cineole. To this end, the German version of the RhinoQoL, a questionnaire measuring symptom frequency, bothersomeness, and impact scales, was applied as a patient-reported outcome measurement.

### Trial Registration

The study was registered on the ClinicalTrials.gov public website with the following identifier: NCT04703673.

## 2. Materials and Methods

### 2.1. Study Setting

This prospective, non-interventional survey was based on recruitment of participants in German pharmacies. Adults suffering from acute or chronic rhinosinusitis (with or without symptoms of bronchitis) who visited a pharmacy for consultation were advised by the pharmacists regarding treatment options. Upon the participant’s decision to administer cineole capsules (Sinolpan) or apply a decongestant nasal spray, their interest in participating in the survey was ascertained. Following the agreement to participate, a questionnaire was handed out to participants for completion (via print out or online via QR code). The survey planned to enrol 500 participants administered cineole capsules and 50 participants applying a decongesting nasal spray (10:1 ratio).

### 2.2. Study Treatments

Upon the participants’ decision to treat their rhinosinusitis symptoms with either cineole capsules (cineole group) or a commercially available decongesting nasal spray (nasal spray group), treatments had to be used in accordance with the respective instruction for use. Cineole treatment allows a maximum daily dose of up to 8 capsules of the Sinolpan product or up to 4 capsules of the Sinolpan forte product (both corresponding to 800 mg cineole).

### 2.3. Questionnaires

Participants were asked to complete a questionnaire prior to the first application of their chosen treatment and at the end of their treatment (or at the latest 10 days after the start of treatment). The first part of the questionnaire (part 1) covered general items, quality of life items, and bronchitis symptoms if applicable. The second part of the questionnaire (part 2) covered post-treatment items and quality of life items. Aside from loss of smell/taste, the questionnaire reflected the characteristics of rhinosinusitis provided by the EPOS guidelines [12].

#### 2.3.1. General Items (Part 1 of Questionnaire)

Prior to the first application of the treatment, participants had to document demographics (sex and age range), indication (rhinosinusitis), and duration of symptoms (days). In addition, severity of disease (1 = slight, 2 = moderate, or 3 = severe), pre-treatment(s) (none, decongesting nasal spray, cough medication, pain killer, or other), chosen treatment (Sinolpan (forte) or name of nasal spray), and concomitant medications (none, decongesting nasal spray, cough medication, pain killer, or other) were documented.

#### 2.3.2. Quality of Life Parameters (Parts 1 and 2 of Questionnaire)

Evaluation of quality of life was assessed with the German version of the validated Rhinosinusitis Quality of Life (RhinQol) questionnaire [13,14,15]. The questionnaire covers symptom frequency, symptom bothersomeness, and symptom impact.

Frequencies were assessed for five symptoms (sinus headache/facial pain/facial pressure, blocked or stuffy nose, postnasal drip, thick nasal discharge, and runny nose) by scoring them from 0 (never) to 4 (always). Mean single (min = 0, max = 4) and sum scores (min = 0, max = 20) were calculated.

Bothersomeness was assessed for three symptoms (sinus headache/facial pain/facial pressure, blocked or stuffy nose, and postnasal drip) and scored from 0 (not troublesome) to 10 (very troublesome). Mean single (min = 0, max = 10) and sum scores (min = 0, maximum = 30) were calculated.

The impact of the disease as indicated by nine items (fatigue, trouble sleeping, concentration problems, performance of normal activities, embarrassment due to nasal symptoms, being frustrated, irritability, sadness, and thoughts on nasal symptoms) was scored from 0 (never) to 4 (always). Mean single (min = 0, max = 4) and sum scores (min = 0, maximum = 36) were calculated.

#### 2.3.3. Bronchitis Symptoms (Parts 1 and 2 of Questionnaire)

Participants also suffering from symptoms of bronchitis at the start of the survey were asked to document five relevant symptoms (cough, sputum, rale, chest pain when coughing, and dyspnoea) prior to the first and after the last application of the study treatment, respectively, after 10 days. Symptoms were scored from 0 (not present) to 4 (very severe). Mean single (min = 0, max = 4) and sum scores (min = 0, maximum = 20) were calculated as described by Lehrl et al. [16].

#### 2.3.4. Post-Treatment Items (Part 2 of Questionnaire)

At the end of treatment (but no longer than 10 days after treatment start), participants were asked to document the treatment duration (days). In addition, the overall efficacy and the overall tolerability were evaluated on a scale from 1 (very good) to 4 (not satisfactory). Furthermore, participants reported changes in quality of life outcomes in relation to work and leisure on a scale from 1 (much better) to 7 (much worse). Finally, participants who were administered cineole were asked if they would recommend the therapy.

#### 2.3.5. Safety

At the start of the survey (part 1 questionnaire), participants were asked to report possible side effects observed during the survey. At the end of the treatment (part 2 questionnaire), participants had to document the presence/absence of side effects and their reporting if applicable.

#### 2.3.6. Statistics

Continuous variables were analysed descriptively (number, mean, standard deviation, median, minimum, maximum, and missing values). A 95% confidence interval was applied. All analyses were performed with IBM SPSS Statistics for Windows, Version 25.0 or higher (Armonk, NY, USA: IBM Corp.). Comparisons between treatment groups were carried out with Mann–Whitney U-test and Wilcoxon test, and *p*-values < 0.05 were considered as statistically significant.

## 3. Results

### 3.1. Study Populations

#### 3.1.1. Disposition and Treatment

Overall, 350 subjects participated in the survey. A total of 310 subjects (88.6%) were administered cineole capsules; of those, the majority used a 200 mg formulation (293, 94.5%), whereas 15 participants (4.8%) used a 100 mg formulation. In addition, 40 subjects (11.4%) applied primarily a decongesting nasal spray. (For a list of nasal spray products, see Appendix A.)

The mean duration of treatment was comparable between groups, with 7.04 ± 2.70 days in the cineole group and 7.30 ± 2.10 days in the nasal spray group.

#### 3.1.2. Baseline Characteristics

The sex distribution was comparable between groups (see Appendix A): there were 196 (63.2%) female subjects within the cineole group and 21 (52.5%) female subjects in the nasal spray group. The age distribution of participants is shown in the Appendix A.

### 3.2. Symptoms, Concomitant Medication, and Quality of Life

#### 3.2.1. Symptomology at Enrolment

Prior to treatment, subjects of the cineole group had suffered from rhinosinusitis symptoms for 5.44 ± 7.48 days compared to 3.95 ± 2.11 days in the nasal spray group. The severity of symptoms preceding the enrolment was evaluated as “moderate” in both groups (1.96 ± 0.60 in the cineole group and 1.83 ± 0.55 in the nasal spray group). Among the participants, one subject of the cineole group stated to have suffered from symptoms for more than 12 weeks, indicating a chronic rhinosinusitis. The remaining participants presented with acute rhinosinusitis. The determination of acute or chronic rhinosinusitis was based on the answers given in the questionnaire. Based on the design of the current survey, a doctor’s visit to confirm the diagnosis was not intended.

#### 3.2.2. Concomitant Medication Prior to and during Treatment

The percentage of participants who treated themselves with medications prior to the start of the study was comparable between groups. In the cineole group, 166 (53.5%) participants were administered other medications prior to treatment, while 142 (44.8%) participants were administered no medications prior to treatment (information was missing for 2 (0.6%) participants). In the nasal spray group, 19 (47.5%) participants were administered other medications prior to treatment, and 20 (50.0%) participants were administered no medications prior to treatment (information was missing for 1 (2.5%) participant).

The type of prior medications differed between the treatment groups. Nasal decongestants were the most frequently used prior concomitant medication in the cineole group (135 (43.5%)), whereas other medications were used most frequently by the nasal spray group (13 (32.5%)) prior to enrolment.

During the treatment period, the percentage of participants using concomitant medication was comparable between the group administered cineole (167 (53.9%)) and the group applying nasal spray (19 (47.5%)). Nasal spray/drops were the most frequently used concomitant medication in the cineole group (135 (43.5%)), whereas the nasal spray group used medications other than nasal spray/drops most frequently as the concomitant agents (15 (37.5%)).

#### 3.2.3. RhinoQol

The mean frequency of all five assessed symptoms decreased significantly (*p* < 0.001 each) upon treatment with cineole capsules (see Figure 1a). Specifically, symptoms including sinus headache/facial pain/facial pressure, postnasal drip, thick nasal discharge, and runny nose were noted “sometimes” at the start of the treatment and “rarely” at the end of the treatment. The symptom of a blocked or stuffy nose was present “often” at the start of the treatment and “rarely” at the end of the treatment. (For single scores, see Appendix A.) The sum of symptom frequencies in the cineole group was 10.18 ± 3.67 prior to commencing treatment and subsequently decreased by 64.0% to 3.67 ± 3.12 post treatment and up until the second assessment (see Figure 2). In comparison, the frequency sum values of the nasal spray treatment group decreased by 55.8% from 9.23 ± 2.94 to 4.08 ± 3.06. (For single scores, see Appendix A.)

The mean bothersomeness of the three symptoms of sinus headache/facial pain/facial pressure, blocked or stuffy nose, and postnasal drip decreased significantly (*p* < 0.001 each) in the cineole group throughout treatment (see Figure 1b). The bothersomeness of sinus headache/facial pain/facial pressure was evaluated with a score of 6.03 ± 2.84 prior to starting treatment with cineole capsules compared to 2.88 ± 2.89 following treatment. The bothersomeness of a blocked or stuffy nose was scored at 6.80 ± 2.45 prior to cineole treatment and decreased to 3.21 ± 2.89. The bothersomeness of postnasal drip decreased from starting values of 4.26 ± 3.11 to end values of 2.08 ± 2.69. The overall sum of the bothersomeness of the three symptoms decreased by 52.1%, i.e., from 16.94 ± 6.94 to 8.11 ± 7.74, in the cineole group and by 39.4%, i.e., from 13.70 ± 6.68 to 8.31 ± 6.45, in the nasal spray group (see Figure 2).

At the start of the treatment, the subjects of the cineole group were affected rarely or sometimes by all nine assessed symptoms (see Appendix A). Upon treatment, the impact of the disease on all assessed symptoms decreased significantly (*p* < 0.001; see Figure 1c). In the cineole group, the sum score of the impact decreased by 53.9%, i.e., from 13.31 ± 6.78 to 6.13 ± 5.71, whereas the sum score in the nasal spray group decreased by 45.3%, i.e., from 10.38 ± 6.95 to 5.68 ± 5.37 (see Figure 2).

The greatest-impacting rhinosinusitis symptoms were assessed to be fatigue, trouble sleeping, and concentration problems. The impact on fatigue decreased by 47.7%, i.e., from 2.19 ± 0.94 to 1.15 ± 0.89; the impact on trouble sleeping decreased by 49.0%, i.e., from 2.08 ± 1.07 to 1.06 ± 0.93; and the impact on concentration problems decreased by 52.0%, i.e., from 1.93 ± 0.99 to 0.92 ± 0.84.

Overall, improvements from start to the end of the survey regarding frequency and impact of rhinosinusitis and symptoms were significantly stronger in the cineole group compared to the nasal spray group (*p* = 0.037 and *p* = 0.028; see Figure 2). Despite not showing a statistically significant difference, when comparing the bothersomeness of symptoms in each group, there was a positive trend towards better outcomes in the cineole group (*p* = 0.061; see Figure 2).

#### 3.2.4. Bronchitis Symptoms

Of the 310 subjects in the cineole group, 68 (21.9%) reported the presence of bronchitis symptoms at the start of the survey. All five bronchitis symptoms (cough, sputum, rale, chest pain when coughing, and dyspnoea) improved significantly (*p* < 0.001) upon treatment. Namely, the mean score of cough decreased from 2.25 ± 0.88 to 1.00 ± 0.68, the mean score of sputum from 1.72 ± 1.06 to 0.56 ± 0.68, the mean score of rale from 1.13 ± 1.15 to 0.23 ± 0.49, the mean score of chest pain from 1.51 ± 1.06 to 0.37 ± 0.60, and the mean score of dyspnoea from 0.76 ± 1.06 to 0.2 ± 0.50 (see Figure 3). The bronchitis sum score decreased significantly (*p* < 0.001) by 69.4%, i.e., from 7.22 ± 4.30 to 2.31 ± 2.35.

#### 3.2.5. Overall Efficacy Evaluation

The overall efficacy of cineole treatment was assessed as good or very good by the majority (279 (90.0%)) of participants, whereas 30 participants (9.7%) evaluated the efficacy as satisfactory (1 missing entry). The efficacy of the nasal spray was assessed as good or very good by 25 (62.5%) participants, whereas 14 participants (35.0%) assessed the treatment as satisfactory and 1 participant (2.5%) as not satisfactory.

#### 3.2.6. Change in Quality of Life Regarding Leisure Time and Work

In both treatment groups, the change in quality of life regarding leisure time and work was assessed at the end of the treatment. The cineole group evaluated both aspects as better: leisure time (2.04 ± 0.93) and work (2.02 ± 0.96). Similarly (*p* = 0.284 and *p* = 0.112), participants of the nasal spray group evaluated quality of life also as better in both aspects: leisure time (2.20 ± 0.76) and work (2.28 ± 0.91).

### 3.3. Safety

#### 3.3.1. Side Effects

Overall, 303 participants (97.7%) of the cineole group reported no possible side effects, whereas 4 participants (1.3%) reported six side effects, and 3 participants (1.0%) did not provide information regarding possible side effects. Participants of the nasal spray group reported no possible side effects. None of the side effects were evaluated as serious, and all side effects were assessed as possibly related to the treatment (Table 1).

#### 3.3.2. Overall Tolerability Evaluation and Recommendation

Within the cineole group, 291 participants (93.9%) evaluated the tolerability of the treatment as good or very good, whereas 14 participants (4.5%) evaluated the tolerability as satisfactory and 1 (0.3%) as not satisfactory, while 4 participants (1.3%) did not evaluate the tolerability. In comparison, 29 participants (72.5%) of the nasal spray group evaluated the tolerability of the treatment as good or very good, whereas 10 participants (25.0%) assessed the tolerability as satisfactory and 1 participant (2.5%) as not satisfactory.

Most participants of the cineole group (277 (89.4%)) would further recommend this treatment, 19 participants (6.1%) would not recommend the product, and 14 participants (4.5%) did not reply to this question.

## 4. Discussion

The current survey aimed to investigate effects of cineole treatment on quality of life in rhinosinusitis patients. The German version of the validated RhinoQoL questionnaire [13,15] was used as a patient-reported outcome measurement. In rhinology, questionnaires are widely used to assess the health-related quality of life, and it has been demonstrated that they are valid measurements for judgement of the burden of the disease for the patient [17].

This was confirmed by the results of the current survey, which demonstrated a significant improvement of rhinosinusitis symptom in the scales of frequency, bothersomeness, and impact upon treatment with cineole. Overall, improvements were more pronounced in subjects administered cineole compared to those applying primarily a decongesting nasal spray. While taking into consideration the small sample size of participants applying nasal spray, the results nevertheless suggest that treatment of rhinosinusitis symptoms with cineole may be more efficient than treatment with a decongesting nasal spray.

A recent meta-analysis investigating the role of herbal medicines in rhinosinusitis supports the findings of this survey, as cineole was found to be among the most effective treatments for acute viral rhinosinusitis [18]. Similarly, another systematic review confirmed positive results using cineole in rhinosinusitis [19]. Both publications draw conclusions from two studies that included fewer participants than enrolled in this survey.

In general, decongestant nasal sprays are recommended to increase nasal breathing and reduce swelling of the ostia of the paranasal sinuses [5]. Of note, there is currently little evidence regarding the clinical relevance of nasal decongestants in rhinosinusitis, and further clinical data are needed to evaluate their beneficial effects [20]. Interestingly, a large proportion of the cineole group (135 (43.5%)) of the current survey concomitantly used nasal decongestants. While the reduction of mucus-production and anti-inflammatory actions account for the clinical benefits of cineole [8,21], decongesting nasal sprays act mainly symptomatically. Therefore, the current results suggest that the combined administration of cineole and a nasal decongestant may provide further therapeutic benefits as compared to therapy with a nasal decongestant alone.

The mean treatment period in the current survey was seven days, which was preceded by the presence of symptoms for four to five days. It should be mentioned that the natural recovery time of rhinosinusitis may vary, and the length of the disease is mainly defined by its classification into acute or chronic forms. Considering that rhinosinusitis is often self-limiting, this survey cannot distinguish between the natural recovery time and the potential acceleration of recovery time attributed to cineole. A review of 15 trials (n = 3057 participants) investigating acute rhinosinusitis in adults showed that 46% of participants were cured without antibiotics (placebo or no treatment) after one week and 64% after 14 days [22]. Another study reported that 50% of rhinosinusitis patients recovered after 5–7 days, and 75% recovered after 8–12 days, with the recovery being negatively associated with a general feeling of illness and reduced productivity [23]. A future study investigating the difference in recovery time of rhinosinusitis patients using cineole compared to its natural recovery time would be desirable.

Bronchitis symptoms in the current survey were only evaluated by a subset of participants (n = 68) administered cineole, therefore limiting their informative value. Nevertheless, a significant improvement (*p* < 0.001) of all assessed bronchitis symptoms from moderate (cough, sputum, and chest pain when coughing) or slight severity (rale, dyspnoea, cough, and sputum) to absence (rale, chest pain when coughing, and dyspnoea) after cineole treatment was demonstrated. Of note, the observed decrease of the BSS (Bronchitis Severity Score) after cineole treatment from 7.22 ± 4.30 to 2.31 ± 2.35 is comparable to the decrease from 7.7 points to 1.4 points described in another non-interventional study [24]. Thus, the current results support the use of cineole as a bronchitis treatment, as shown in other trials [11].

Seeing that only one participant of this survey suffered from chronic rhinosinusitis, the found results should mainly be seen in the context of acute rhinosinusitis. Further research involving cineole in chronic rhinosinusitis patients would be useful. 

A limitation of the current survey concerns the number of patients in the nasal spray group (n = 50). Therefore, conclusions regarding the possible therapeutic benefit of using the combination of cineole and a nasal decongestant in contrast to using a nasal decongestant without cineole should be interpreted with caution. As the non-interventional design of this work limits the significance of the findings, a randomized controlled trial is necessary to explore this hypothesis further. While realising that the use of concomitant medication according to the individual patient’s needs is a relevant limitation of this study, this also reflects the genuine real-world situation, which can almost exclusively be displayed and analysed with this kind of study and therefore provides valuable insights. Despite the described limitations, the high number of participants in the cineole group reflected in the presented results add valuable information to scientific knowledge regarding cineole in rhinosinusitis treatment.

Taken together, cineole treatment of rhinosinusitis was well tolerated and resulted in clear quality of life improvements. It may be a useful concomitant treatment option supplementing standard rhinosinusitis therapies.

## Figures and Tables

**Figure 1 medicines-10-00037-f001:**
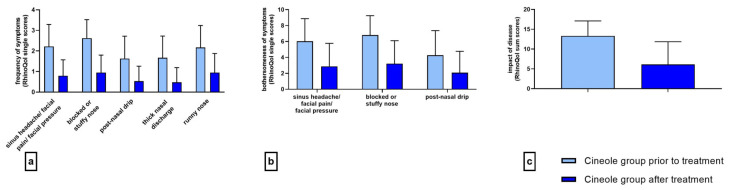
Frequency ((**a**) 5-point scale), bothersomeness ((**b**) 11-point scale), and impact ((**c**) 37-point sum scale) of rhinosinusitis symptoms prior to and after treatment with cineole capsules. Data are presented as mean +SD.

**Figure 2 medicines-10-00037-f002:**
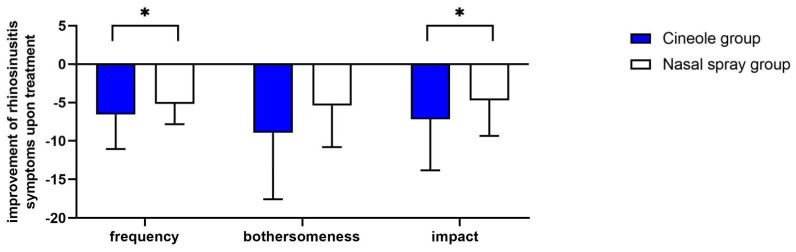
Decreases in sum scores of the frequency, bothersomeness, and impact of rhinosinusitis symptoms upon treatment with cineole capsules or nasal spray. Data are presented as mean + SD. * *p* < 0.05.

**Figure 3 medicines-10-00037-f003:**
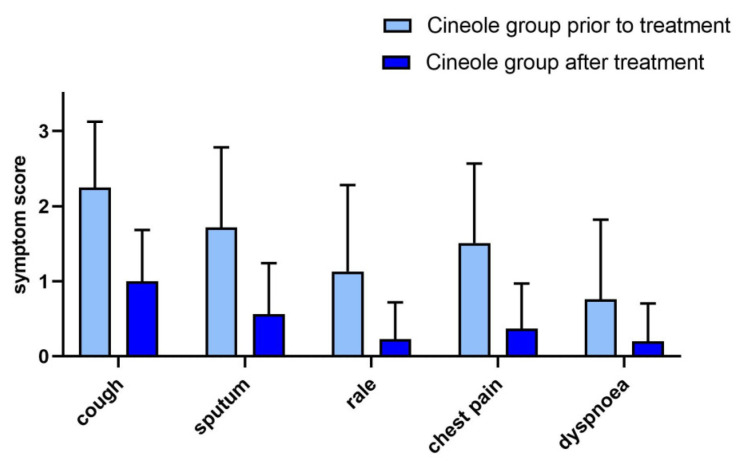
Bronchitis symptoms (5-point scale) prior to and after treatment with cineole capsules. Data are presented as mean + SD.

**Table 1 medicines-10-00037-t001:** List of side effects reported by participants administered cineole capsules during the survey.

Participant ID	Side Effect (LLT Code)	Relationship to Treatment	Outcome
026-1	General discomfort	Possibly related	Recovered
026-1	Stomach, burning sensation	Possibly related	Recovered
072-1	Nausea	Possibly related	Unknown
072-3	Taste altered	Possibly related	Unknown
196-2	Gastrointestinal cramps	Possibly related	Unknown
196-2	Pain	Possibly related	Unknown

## Data Availability

The data presented in this manuscript are available on reasonable request from the sponsor.

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
