# Peer review of "Rhinosinusitis Treatment with Cineole: Patient-Reported Quality of Life Improvements from a Non-Interventional, Pharmacy-Based Survey"

_medicines, 2023, doi:10.3390/medicines10060037_

Round 1
Reviewer 1 Report
Dear colleagues,
In this manuscript, the authors tried to investigate the quality of life in participants suffering from rhinosinusitis using Cineole by performing cohorts study with 500 participants administering Cineole capsules and 50 participants applying a decongesting nasal spray (10:1 ratio). Despite a good impression of the article, I have some remarks and questions which could improve the article in my opinion.
What kinds of patients were included in this research?
Which type of rhinosinusitis did patients have according to the EPOS classification?
From the manuscript, I could suspect that these patients had acute viral rhinosinusitis. If it is so, would you please clarify how did you exclude patients who suffered from chronic rhinosinusitis with or without nasal polyps?
Did you exclude also patients who suffered from allergic rhinitis?
Using of “quality of life” as a keyword isn't too clear in that article, so I suggest replacing it for another. “Eucalyptus oil” as keyword could be added in my opinion.
The authors said about “Data is presented as mean ±SD” in the signature to figures 1, 2, 3. But only +SD is presented in diagrams.
In summary, I have been satisfied with the level of the article. I hope this manuscript will attract significant attention from the research community. In my personal opinion, the article is valuable for publication in “Medicines” after the explanation of named above points and correction.
Reviewer 2 Report
In the last few years there has been much interest in cineole, and other phytopharmaceuticals, which seem to have an interesting role in controlling the symptoms of upper airway pathologies and beyond (many articles investigate the usefulness of these molecules in lung pathologies). This article highlights concepts already present in the literature, however, given the good number of participants, the work is remarkable because it has a greater statistical value than others studies that take into consideration the same molecule (alone). In fact, the literature includes more studies in which mixes of phytotherapeutics are taken into consideration.
Nowadays, evidence in literature that any herbal medicines are beneficial in the treatment of rhinosinusitis is limited, particularly in chronic rhinosinusitis1.
The aim of this paper is to observe and organize data collected from a validated quastionnaire, without any type of intervention in the process of administration of the drugs, to see changes in patients quality of life. The evaluation, in the end, seem to be a pure analisys.
The article is well-structured and clear. The objective is explicit right from the title, it is a survey, and the authors try to enlight the pure results of data collected. The study does not investigate secondary outcomes. Despite this, it is appreciable the attention to the elements that have arisen during the work, for example the table of the side effects encountered -Table 1.- is immedialetly available and useful).
The introduction (lines 40-41) also refers to a study that highlighted the anti-inflammatory capabilities of cineole from a biochemical point of view. Sometimes molecular strategies and mechanisms of action about some phytotherapeutics remain obscure. Having awareness of the biochemical mechanism of Cineole is certainly an excellent prerequisite for deepening the clinical benefits of this molecule.
-In the literature there is a very recent meta-analysis4, dating back to January 2023, which summarize what the literature knows about the role of the various herbal medicines as a treatment for rhinosinusitis. This meta-analysis includes two studies2,3 on cineole (also cited in the reviewed article) both produced in Germany. The data obtained from these two ancillary studies regarding Cineole is diluted within the aforementioned meta-analysis, which investigates the existing literature on the entire range of phytotherapeutics available today.
Although the meta-analysis reports among its conclusions that “Cineole improved patient important outcomes” and “The most beneficial agents for symptom and HRQoL improvements in acute post-viral rhinosinusitis were Cineole and Spicae aetheroleum, respectively” these conclusions are drawn from two studies (randomized trials) dated and with a lower number of patients enrolled.-
Most of the articles cited in the construction of the study are dated, however due to the scarcity of studies in the literature on the clinical use of the cineole (alone) this data is understandable and does not detract value from the study.
The study is designed according to the authors' intent, and the hypothesis at the beginning of the study is respected during the process. Furthermore, the steps illustrated in the materials and methods section are extensively detailed and sufficient to make the study reproducible in other settings. Figures and tables are easy to understand and offer a quick and very direct graphical representation of the results obtained from the comparison between cineoles and others nasal sprays.
The statistical methods used are simple to understand but well structured, there seems to be no flaws in their construction. The results of the analyzes are illustrated with a confidence interval, p<0.05 is taken into consideration. The results appear to be statistically significant for almost all parts of the survey and therefore enjoy a reasonable statistical strength.
In conclusion: methods are sufficiently described; the software used by the authors to produce the analyzes is illustrated; the work is adapted to the standards of the scientific community.
Moreover, the article is able to illustrate its limitations:
- 1) the concurrence of several treatments during the intake of cineol or nasal decongestants is mentioned, argued and analyzed (the percentages of sub-population exposed to other products possibly "contaminating" the results of the study are reported). (paragraph 3.2.2. Lines: 158 to 175), However it must be said that the percentages of subpopulation between the two groups are almost similar.
- 2) it is specified that the natural course of rhinitis is often self-limiting and short: there could be a bias due to the overlapping of the therapy with the natural shutdown of the nasosinus inflammation process. (4. Discussion. Lines 289 to 299)
In the end, the work does not identify a gap in knowledge since the two studies cited before shown similar results, but the study is relevant in his field because it offer a stimulant assumption for the scientific community to continue the research about the effects, clinical benefits of this molecule.
References
1 Guo R, Canter PH, Ernst E. Herbal medicines for the treatment of rhinosinusitis: a systematic review. Otolaryngol Head Neck Surg. 2006 Oct;135(4):496-506. doi: 10.1016/j.otohns.2006.06.1254. PMID: 17011407.
2 Kehrl W, Sonnemann U, Dethlefsen U. Therapy for acute nonpurulent rhinosinusitis with cineole: results of a double-blind, randomized, placebo-controlled trial. Laryngoscope. 2004 Apr;114(4):738-42. doi: 10.1097/00005537-200404000-00027. PMID: 15064633.
3 Tesche S, Metternich F, Sonnemann U, Engelke JC, Dethlefsen U. The value of herbal medicines in the treatment of acute non-purulent rhinosinusitis. Results of a double-blind, randomised, controlled trial. Eur Arch Otorhinolaryngol. 2008 Nov;265(11):1355-9. doi: 10.1007/s00405-008-0683-z. Epub 2008 Apr 25. PMID: 18437408.
4 Hoang MP, Seresirikachorn K, Chitsuthipakorn W, Snidvongs K. Herbal Medicines for Rhinosinusitis: A Systematic Review and Network Meta-analysis. Curr Allergy Asthma Rep. 2023 Feb;23(2):93-109. doi: 10.1007/s11882-022-01060-z. Epub 2023 Jan 7. PMID: 36609950.
